# LARGE SCALE GRAPH LEARNING FROM SMOOTH SIGNALS

**Vassilis Kalofolias**
Signal Processing Laboratory 2, EPFL
Station 11, 1015 Lausanne, Switzerland
v.kalofolias@gmail.com

**Nathanaël Perraudin**
Swiss Data Science Center, ETH Zürich
Universitätstrasse 25, 8006 Zürich, Switzerland
nperraud@ethz.ch

## ABSTRACT

Graphs are a prevaluable tool in data science, as they model the inherent structure of the data. Typically they are constructed either by connecting nearest samples, or by learning them from data, solving an optimization problem. While graph learning does achieve a better quality, it also comes with a higher computational cost. In particular, the current state-of-the-art model cost is $\mathcal{O}\left(n^2\right)$ for $n$ samples. In this paper, we show how to scale it, obtaining an approximation with leading cost of $\mathcal{O}\left(n \log(n)\right)$, with quality that approaches the exact graph learning model. Our algorithm uses known approximate nearest neighbor techniques to reduce the number of variables, and automatically selects the correct parameters of the model, requiring a single intuitive input: the desired edge density.

## 1 INTRODUCTION

Graphs are an invaluable tool in data science, as they can capture complex structures inherent in seemingly irregular high-dimensional data. While classical applications of graphs include data embedding, manifold learning, clustering and semi-supervised learning (Zhu et al., 2003; Belkin et al., 2006; Von Luxburg, 2007), they were later used for regularizing various machine learning models, for example for classification, sparse coding, matrix completion, or PCA (Zhang et al., 2006; Zheng et al., 2011; Kalofolias et al., 2014; Shahid et al., 2016).

More recently, graphs drew the attention of the deep learning community. While convolutional neural networks (CNNs) were highly successful for learning image representations, it was not obvious how to generalize them for irregular high dimensional domains, were standard convolution is not applicable. Graphs bridge the gap between irregular data and CNNs through the generalization of convolutions on graphs (Defferrard et al., 2016; Kipf & Welling, 2016; Monti et al., 2016; Li et al., 2015). While clearly graph quality is important in such applications (Henaff et al., 2015; Defferrard et al., 2016), the question of how to optimally construct a graph remains an open problem.

The first applications mostly used weighted $k$-nearest neighbors graphs ($k$-NN) (Zhu et al., 2003; Belkin et al., 2006; Von Luxburg, 2007), but the last few years more sophisticated methods of *learning* graphs from data were proposed. Today, *graph learning*, or *network inference*, has become an important problem itself (Wang & Zhang, 2008; Daitch et al., 2009; Jebara et al., 2009; Lake & Tenenbaum, 2010; Hu et al., 2013; Dong et al., 2015; Kalofolias, 2016).

Hoever, graph learning is computationally too costly for large-scale applications that need graphs between millions of samples. Current state of the art models for learning weighted undirected graphs (Dong et al., 2015; Kalofolias, 2016) cost $\mathcal{O}\left(n^2\right)$ per iteration for $n$ nodes, while previous solutions are even more expensive. Furthermore, they need parameter tuning to control sparsity, and this adds an extra burden making them prohibitive for applications with more than a few thousands of nodes.

Large-scale applications can only resort to approximate nearest neighbors (A-NN), e.g. (Dong et al., 2011; Muja & Lowe, 2014; Malkov & Yashunin, 2016), that run with a cost of $\mathcal{O}\left(n \log(n)\right)$. This is low compared to computing a simple $k$-NN graph, as even the pairwise distance matrix between all samples needs $\mathcal{O}\left(n^2\right)$ computations. However, the quality of A-NN is worse than the $k$-NN, that is already not as good as if we would learn the graph from data.

*In this paper, we propose the first scalable graph learning method, with the same leading cost as A-NN, and with quality that approaches state-of-the-art graph learning.* Our method leverages A-NN

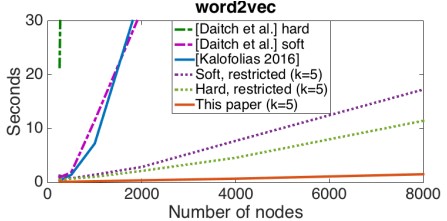 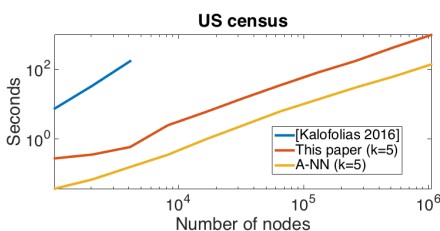

Figure 1: Time comparison of different ways to compute a graph. **Left:** Graph between 10,000 most frequent English words using a word2vec representation. **Right:** Graph between 1,000,000 nodes from 68 features (US Census 1990). Scalable algorithms benefit from a small average node degree $k$.

graphs to effectively reduce the number of variables, and the state-of-the-art graph learning model by Kalofolias (2016) in order to achieve the best of both worlds: low cost and good quality. In Figure 1 we illustrate the advantage of our solution compared to the current state-of-the-art. Note that while the standard model costs the same regardless of the graph density (average node degree) $k$, our solution benefits from the desired graph sparsity to reduce computation.

One of our key contributions is to provide *a method to automatically select the parameters of the model* by Kalofolias (2016) given a desired graph sparsity level. Like in $k$-NN, the user can choose the number of neighbors $k$, *without performing grid search* over two parameters. Using our scheme, we can learn a 1-million-nodes graph with a desired sparsity level on a desktop computer in 16 minutes, with a simple Matlab implementation.

## 2 GRAPH LEARNING FROM SMOOTH SIGNALS

A widely used assumption for data residing on graphs is that values change smoothly across adjacent nodes. The smoothness of a set of vectors $x_1, \ldots, x_n \in \mathbb{R}^d$ on a given weighted undirected graph is usually quantified by the *Dirichlet energy* (Belkin & Niyogi, 2001)

$$\frac{1}{2} \sum_{i,j} W_{ij} \|x_i - x_j\|^2 = \operatorname{tr}\left(X^\top L X\right), \tag{1}$$

where $W_{ij} \in \mathbb{R}_+$ denotes the weight of the edge between nodes $i$ and $j$, $L = D - W$ is the graph Laplacian, and $D_{ii} = \sum_j W_{ij}$ is the diagonal weighted degree matrix.

The first works for graph learning focused on learning the weights of a fixed k-nearest neighbor pattern (Wang & Zhang, 2008), learning a binary pattern (Jebara et al., 2009) or the whole adjacency matrix (Daitch et al., 2009). A more recent family of models is based on minimizing the *Dirichlet energy* on a graph (Lake & Tenenbaum, 2010; Hu et al., 2013; Dong et al., 2015; Kalofolias, 2016). In the latter, Kalofolias (2016) proposed a unified model for learning a graph from smooth signals, that reads as follows:

$$\min_{W \in \mathcal{W}} \|W \circ Z\|_{1,1} + f(W). \tag{2}$$

Here, $Z_{ij} = \|x_i - x_j\|^2$, $\circ$ denotes the Hadamard product, and the first term is equal to $\operatorname{tr}\left(X^\top L X\right)$. The optimization is over the set $\mathcal{W}$ of valid adjacency matrices (non-negative, symmetric, with zero diagonal).

The role of matrix function $f(W)$ is to prevent $W$ from obtaining a trivial zero value, control sparsity, and impose further structure, depending on the data and the application. Kalofolias obtained state-of-the-art results using

$$f(W) = -\alpha \mathbf{1}^\top \log(W\mathbf{1}) + \frac{\beta}{2} \|W\|_F^2, \tag{3}$$

where $\mathbf{1} = [1, \ldots 1]^\top$. We will call this the *log model*. The previous state of the art was proposed by Hu et al. (2013) and by Dong et al. (2015), using

$$f(W) = \alpha \|W\mathbf{1}\|^2 + \alpha \|W\|_F^2 + \mathbb{1}\left\{\|W\|_{1,1} = n\right\}, \tag{4}$$

where $\mathbb{1}\{\text{condition}\} = 0$ if condition holds, $\infty$ otherwise. In the sequel we call this the $\ell_2$ *model*. Since $W\mathbf{1}$ is the node degrees' vector, the *log* model (3) prevents the formation of disconnected nodes due to the logarithmic barrier, while the $\ell_2$ model (4) controls sparsity by penalizing large degrees due to the first term.

## 3 CONSTRAINED EDGE PATTERN

In traditional graph learning, all $\binom{n}{2}$ possible edges between $n$ nodes are considered, that results in a cost of at least $\mathcal{O}\left(n^2\right)$ computations per iteration. Often, however, we need graphs with a roughly fixed number of edges per node, like in $k$-NN graphs. It is natural to then ask ourselves whether the cost of graph learning can be reduced, reflecting the final desired graph sparsity.

In fact, the original problem (2) for the log model (3) can be solved efficiently when a constrained set $\mathcal{E}^{\text{allowed}} \subseteq \{(i,j) : i < j\}$ of allowed edges is known a priori. In that case, it suffices to solve the modified problem

$$\underset{W \in \widetilde{\mathcal{W}}}{\text{minimize}} \; \|W \circ Z\|_{1,1} \; - \; \alpha \mathbf{1}^\top \log(W\mathbf{1}) \; + \; \frac{\beta}{2}\|W\|_F^2, \tag{5}$$

where we optimize in the constrained set of adjacency matrices $W \in \widetilde{\mathcal{W}}$. After reducing the set of edges to $\mathcal{E}^{\text{allowed}}$, it suffices to solve the modified problem (5) Following Kalofolias (2016), we can rewrite the problem as

$$\underset{\widetilde{w}}{\text{minimize}} \; f_1(\widetilde{w}) + f_2(K\widetilde{w}) + f_3(\widetilde{w}), \tag{6}$$

with

$$f_1(\widetilde{w}) = \mathbb{1}\{\tilde{w} \geq 0\} + 2\tilde{w}^\top \tilde{z},$$
$$f_2(v) = -\alpha \mathbf{1}^\top \log(v),$$
$$f_3(\widetilde{w}) = \beta\|\widetilde{w}\|^2, \text{ with } \zeta = 2\beta,$$

where $\zeta$ is the Lipschitz constant of $f_3$. Note that we gather all free parameters of the adjacency matrix $\widetilde{W} \in \widetilde{\mathcal{W}}_m$ in a vector $\widetilde{w} \in \widetilde{\mathcal{W}}_v$ of size only $|\mathcal{E}^{\text{allowed}}|$, that is, the number of allowed edges, each counted only once. Accordingly, in $\tilde{z} = z(\mathcal{E}^{\text{allowed}})$ we only keep the corresponding pairwise distances from matrix $Z$. The linear operator $K = \widetilde{S} = S(:, \mathcal{E}^{\text{allowed}})$ is also modified, keeping only the columns corresponding to the edges in $\mathcal{E}^{\text{allowed}}$.

In this form, the problem can be solved by the primal dual techniques by Komodakis & Pesquet (2015). The cost of the dual step, operating on the dual variable $v$ (degrees vector) remains $\mathcal{O}(n)$. However, the cost of the primal step, as well as the cost of applying the modified operator $\widetilde{S}$ in order to exchange between the primal and dual spaces is $\mathcal{O}\left(\mathcal{E}^{\text{allowed}}\right)$ instead of $\mathcal{O}\left(n^2\right)$ of the initial algorithm 1 by Kalofolias (2016), reducing the overall complexity.

In some cases, a pattern of allowed edges $\mathcal{E}^{\text{allowed}}$ can be induced by constraints of the model, for example sensor networks only assume connections between geographically nearby sensors. In most applications, however, a constrained set is not known beforehand, and we need to approximate the edge support of the final learned graph in order to reduce the number of variables. To this end, we propose using approximate nearest neighbors graphs to obtain a good approximation. While computing a $k$-NN graph needs $\mathcal{O}\left(n^2d\right)$ computations, *approximate nearest neighbors* (A-NN) algorithms (Muja & Lowe, 2009; Dong et al., 2011; Muja & Lowe, 2012; 2014; Malkov & Yashunin, 2016) offer a good compromise between accuracy and speed. Specifically, A-NN methods scale gracefully with the number of nodes $n$, the fastest ones having an overall complexity of $\mathcal{O}(n\log(n)d)$ for $d$-dimensional data.

When approximating the support of the final edges of a graph, we prefer to have false positives than false negatives. We thus start with an initial support with a larger cardinality than that of the desired final graph, and let the weight learning step automatically select which edges to set to zero. We select a set $\mathcal{E}^{\text{allowed}}$ with cardinality $|\mathcal{E}^{\text{allowed}}| = \mathcal{O}(nkr)$, where $k$ is the desired number of neighbors per node and $r$ a small multiplicative factor. By setting the sparsity parameters correctly, the graph learning step will only keep the final $\mathcal{O}(nk)$ edges, setting the less important or wrong edges to zero. The bigger the factor $r$, the more freedom the learning algorithm has to select the right edges.

### 3.1 OVERALL THEORETICAL COMPLEXITY

The cost of learning a $kr$-A-NN graph is $\mathcal{O}(n\log(n)d)$ for $n$ nodes and data in $\mathbb{R}^d$, while additionally learning the edge weights costs $\mathcal{O}(krn)$ per iteration. The overall complexity is therefore $\mathcal{O}(n\log(n)d) + \mathcal{O}(nkrI)$ for $I$ iterations. For large $n$, the dominating cost is asymptotically the one of computing the A-NN and not the cost of learning the weights on the reduced set.

## 4 AUTOMATIC PARAMETER SELECTION

A major problem of models (3) and (4) is the choice of meaningful parameters $\alpha, \beta$, as grid search increases computation significantly. We show how this burden can be completely avoided for model (3). First, we show that sparsity depends effectively on a single parameter, and then we propose a method to set it automatically for any $k$. Our method is based on predicting the number of edges of any node for a given parameter value, if we relax the symmetricity constraint of $W$.

### 4.1 REDUCTION TO A SINGLE OPTIMIZATION PARAMETER

In (Kalofolias, 2016, Proposition 2), it is argued that model (3) effectively has one parameter changing the shape of the edges, the other changing the magnitude. We reformulate this claim as follows:

**Proposition 1.** *Let $W^*(Z, \alpha, \beta)$ denote the solution of model (3) for input distances $Z$ and parameters $\alpha, \beta > 0$. Then the same solution can be obtained with fixed parameters $\alpha = 1$ and $\beta = 1$, by multiplying the input distances by $\theta = \frac{1}{\sqrt{\alpha\beta}}$ and the resulting edges by $\delta = \sqrt{\frac{\alpha}{\beta}}$:*

$$W^*(Z, \alpha, \beta) = \sqrt{\frac{\alpha}{\beta}} W^*\left(\frac{1}{\sqrt{\alpha\beta}} Z, 1, 1\right) = \delta W^*(\theta Z, 1, 1). \tag{7}$$

**Proof.** *Apply (Kalofolias, 2016, Prop. 2), with $\gamma = \sqrt{\frac{\alpha}{\beta}}$ and divide all operands by $\sqrt{\alpha\beta}$.*

Proposition 1 shows that the parameter spaces $(\alpha, \beta)$ and $(\theta, \delta)$ are equivalent. While the first one is convenient to define (3), the second one allows independent tuning of sparsity and scale of the solution. For some applications the scale of the graph is less important, and multiplying all edges by the same constant does not change its functionality. In other cases, we want to explicitly normalize the graph to a specific size, for example setting $\|W\|_{1,1} = n$ as in (Hu et al., 2013), or making sure that $\lambda_{\max} = 1$. Nevertheless, for applications where scale shall be set automatically, we provide the following Theorem that characterizes the connection between $\delta$ and the scale of the weights.

**Theorem 1.** *All edges of the solution $W^*(Z, \alpha, \beta) = \delta W^*(\theta Z, 1, 1)$ with $\theta = \frac{1}{\sqrt{\alpha\beta}}$, $\delta = \sqrt{\frac{\alpha}{\beta}}$ of model (3) are upper bounded by $\delta$: $W^*(Z, \alpha, \beta) \leq \sqrt{\frac{\alpha}{\beta}} = \delta$.*

**Proof.** *See supplementary material A.*

### 4.2 SETTING THE REMAINING REGULARIZATION PARAMETER

The last step for automatizing parameter selection is to find a relation between $\theta$ and the desired sparsity (the average number of neighbors per node). We first analyze the sparsity level with respect to $\theta$ for each node independently. Once the independent problems are well characterized, we propose an empirical solution to obtain a global value of $\theta$ providing approximately the desired sparsity level.

#### 4.2.1 SPARSITY ANALYSIS FOR ONE NODE

In order to analyze the sparsity of the graphs obtained by the model (3), we take one step back and drop the symmetricity constraint. The problem becomes separable and we can focus on only one node. Keeping only one column $w$ of matrix $W$, we arrive to the simpler optimization problem

$$\min_{w \in \mathbb{R}^n_+} \quad \theta w^\top z - \log(w^\top \mathbf{1}) + \frac{1}{2}\|w\|_2^2. \tag{8}$$

The above problem also has only one parameter $\theta$ that controls sparsity, so that larger values of $\theta$ yield sparser solutions $w^*$. Furthermore, *it enjoys an analytic solution if we sort the elements of $z$*, as we prove with the next theorem.

**Theorem 2.** *Suppose that the input vector $z$ is sorted in ascending order. Then the solution of problem (8) has the form*

$$w^* = \max(0, \lambda^* - \theta z) = [\lambda^* - \theta z_{\mathcal{I}}; \mathbf{0}], \tag{9}$$

*with*

$$\lambda^* = \frac{\theta b_k + \sqrt{\theta^2 b_k^2 + 4k}}{2k}.$$

*The set $\mathcal{I} = \{1, \ldots, k\}$ corresponds to the indices of the $k$ smallest distances $z_i$ and $b_k$ is the cumulative sum of the smallest $k$ distances in $z$, $b_k = \sum_{i=1}^k z_i$.*

We provide the proof of Theorem 1 after presenting certain intermediate results. In order to solve Problem (8) we first introduce a slack variable $l$ for the inequality constraint, so that the KKT optimality conditions are

$$\theta z - \frac{1}{w^\top \mathbf{1}} + w - l = 0, \tag{10}$$

$$w \geq 0, \tag{11}$$

$$l \geq 0, \tag{12}$$

$$l_i w_i = 0, \forall i. \tag{13}$$

the optimum of $w$ can be revealed by introducing the term $\lambda^* = \frac{1}{w^{*\top}\mathbf{1}}$ and rewrite (10) as

$$w^* = \lambda^* - \theta z + l. \tag{14}$$

Then, we split the elements of $w$ in two sets, $\mathcal{A}$ and $\mathcal{I}$ according to the activity of the inequality constraint (11), so that $w_{\mathcal{I}} > \mathbf{0}$ (inactive) and $w_{\mathcal{A}} = \mathbf{0}$ (active). Note that at the minimum, the elements of $w$ will also be sorted in a descending order so that $w^* = [w_{\mathcal{I}}^*; \mathbf{0}]$, according to Theorem 2. We first need a condition for an element of $w^*$ to be positive, as expressed in the following lemma:

**Lemma 1.** *An element $w_i^*$ of the solution $w^*$ of problem (8) is in the active set $\mathcal{A}$ if and only if it corresponds to an element of $z_i$ for which $\theta z_i \geq \lambda^*$.*

**Proof.** ($\Rightarrow$): *If $w_i$ is in the active set we have $w_i = 0$ and $l_i \geq 0$, therefore from (14) we have $\theta z_i - \lambda^* \geq 0$. ($\Leftarrow$): Suppose that there exists $i \in \mathcal{I}$ for which $\theta z_i \geq \lambda^*$. The constraint being inactive means that $w_i^* > 0$. From (13) we have that $l_i = 0$ and (14) gives $w_i^* = \lambda^* - \theta z_i \leq 0$, a contradiction.*

We are now ready to proceed to the proof of Theorem 2.

**Proof** (Theorem 2). *As elements of $\theta z$ are sorted in an ascending order, the elements of $\lambda^* - \theta z$ will be in a descending order. Furthermore, we know from Lemma 1 that all positive $w_i^*$ will correspond to $\theta z_i < \lambda^*$. Then, supposing that $|\mathcal{I}| = k$ we have the following ordering:*

$$-\theta z_1 \geq \cdots \geq -\theta z_k > -\lambda^* \geq -\theta z_{k+1} \geq \cdots \geq -\theta z_n \Rightarrow$$
$$\lambda^* - \theta z_1 \geq \cdots \geq \lambda^* - \theta z_k > 0 \geq \lambda^* - \theta z_{k+1} \geq \cdots \geq \lambda^* - \theta z_n.$$

*In words, the vector $\lambda^* - \theta z$ will have sorted elements so that the first $k$ are positive and the rest are non-positive. Furthermore, we know that the elements of $l$ in the optimal have to be $0$ for all inactive variables $w_{\mathcal{I}}^*$, therefore $w_{\mathcal{I}}^* = \lambda^* - \theta z_{\mathcal{I}}$. The remaining elements of $w$ will be $0$ by definition of the active set:*

$$w^* = [\underbrace{\lambda^* - \theta z_1, \cdots, \lambda^* - \theta z_k}_{w_{\mathcal{I}}^*}, \underbrace{0, \cdots, 0}_{w_{\mathcal{A}}^*}].$$

*What remains is to find an expression to compute $\lambda^*$ for any given $z$. Keeping $z$ ordered in ascending order, let the cumulative sum of $z_i$ be $b_k = \sum_{i=1}^{k} z_i$. Then, from the definition of $\lambda^* = \frac{1}{w^{*\top}\mathbf{1}}$ and using the structure of $w^*$ we have*

$$w^{*\top}\mathbf{1}\lambda^* = 1 \quad \Rightarrow \quad \left(k\lambda^* - \theta z_{\mathcal{I}}^\top \mathbf{1}\right)\lambda^* = 1 \quad \Rightarrow \quad k(\lambda^*)^2 - \theta b_k \lambda^* - 1 = 0, \tag{15}$$

*which has only one positive solution,*

$$\lambda^* = \frac{\theta b_k + \sqrt{\theta^2 b_k^2 + 4k}}{2k}. \tag{16}$$

### 4.2.2 PARAMETER SELECTION FOR THE NON-SYMMETRIC CASE

While Theorem 2 gives the form of the solution for a known $k$, the latter cannot be known a priori, as it is also a function of $z$. For this, we propose Algorithm 1 that solves this problem, simultaneously finding $k$ and $\lambda^*$ in $\mathcal{O}(k)$ iterations. This algorithm will be needed for automatically setting the parameters for the symmetric case of graph learning.

As $k$ is the number of non-zero edges per node, we can assume it to be small, like for $k$-NN. It is thus cheap to incrementally try all values of $k$ from $k = 1$ until we find the correct one, as Algorithm 1 does. Once we try a value that satisfies $\lambda \in (\theta z_k, \theta z_{k+1}]$, all KKT conditions hold and we have found the solution to our problem. A similar algorithm was proposed in (Duchi et al., 2008)

for projecting vectors on the probability simplex, that could be used for a similar analysis for the $\ell_2$-degree constraints model (4).

Most interestingly, using the form of the solution given by Theorem 2 we can solve the reverse problem: *If we know the distances vector $z$ and we want a solution $w^*$ with exactly $k$ non-zero elements, what should the parameter $\theta$ be?* The following theorem answers this question, giving intervals for $\theta$ as a function of $k$, $z$ and its cumulative sum $b$.

**Theorem 3.** *Let $\theta \in \left( \dfrac{1}{\sqrt{kz_{k+1}^2 - b_k z_{k+1}}}, \dfrac{1}{\sqrt{kz_k^2 - b_k z_k}} \right]$, the solution of eq. (8) has exactly $k$ non-zeros.*

**Proof.** *See supplementary material B.*

The idea of Theorem 3 is illustrated in the left part of Figure 2. For this figure we have used the distances between one image of MNIST and 999 other images. For any given sparsity level $k$ we know what are the intervals of the valid values of $\theta$ by looking at the pairwise distances.

---

**Algorithm 1** Solver of the one-node problem, (8).

1: **Input:** $z \in \mathbb{R}_{*+}^n$ in ascending order, $\theta \in \mathbb{R}_{*+}$
2: $b_0 \leftarrow 0$ {Initialize cumulative sum}
3: **for** $i = 1, \ldots, n$ **do**
4:     $b_i \leftarrow b_{i-1} + z_i$ {Cumulative sum of z}
5:     $\lambda_i \leftarrow \dfrac{\sqrt{\theta^2 b_i^2 + 4i} + \theta b_i}{2i}$
6:     **if** $\lambda_i > \theta z_i$ **then**
7:         $k \leftarrow i - 1$
8:         $\lambda^* \leftarrow \lambda_k$
9:         $w^* \leftarrow \max\{0, \lambda^* - \theta z\}$ {$k$-sparse output}
10:         **break**
11:     **end if**
12: **end for**

---

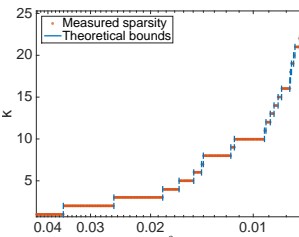 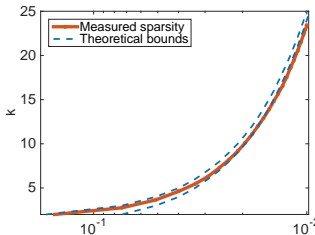 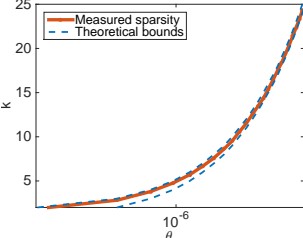

Figure 2: Theoretical bounds of $\theta$ for a given sparsity level on 1000 images from MNIST. **Left**: Solving (8) for only one column of $Z$. Theorem 3 applies and for each $k$ gives the bounds of $\theta$ (blue). **Middle**: Solving (3) for the whole pairwise distance matrix $Z$ of the same dataset. The bounds of (17) (blue dashed line) are used to approximate the sparsity of the solution. The red line is the measured sparsity of the learned graphs from model (3). **Right**: Same for USPS dataset.

### 4.2.3 PARAMETER SELECTION FOR THE SYMMETRIC CASE

In order to approximate the parameter $\theta$ that gives the desired sparsity of $W$, we use the above analysis for each row or column separately, omitting the symmetricity constraint. Then, using the arithmetic mean of the bounds of $\theta$ we obtain a good approximation of the behaviour of the full symmetric problem. In other words, to obtain a graph with approximately $k$ edges per node, we propose to use the following intervals:

$$\theta_k \in \left( \frac{1}{n} \sum_{j=1}^n \theta_{k,j}^{lower}, \frac{1}{n} \sum_{j=1}^n \theta_{k,j}^{upper} \right] = \left( \sum_{j=1}^n \frac{1}{n\sqrt{k\hat{Z}_{k+1,j}^2 - B_{k,j}\hat{Z}_{k+1,i}}}, \sum_{j=1}^n \frac{1}{n\sqrt{k\hat{Z}_{k,j}^2 - B_{k,j}\hat{Z}_{k,j}}} \right], \quad (17)$$

where $\hat{Z}$ is obtained by sorting each column of $Z$ in increasing order, and $B_{k,j} = \sum_{i=1}^k \hat{Z}_{i,j}$. The above expression is the arithmetic mean over all minimum and maximum values of $\theta_{k,j}$ that would give a $k$-sparse result $W_{:,j}$ if we were to solve problem (8) for each of columns separately, according to Theorem 3. Even though the above approach does not take into account the symmetricity constraints, it gives surprisingly good results in the vast majority of the cases. For the final value of $\theta$ for a given $k$, we use the middle of this interval in a logarithmic scale, that is, the *geometric mean* of the upper and lower limits.

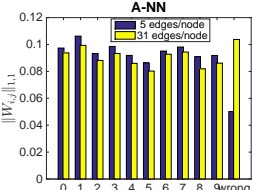 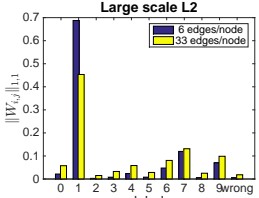 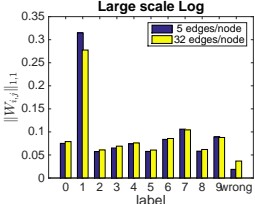

Figure 5: Connectivity across classes of MNIST. **Left**: A-NN graph. **Middle**: $\ell_2$ model (4) neglects digits with larger distance. **Right**: log model (5) does not neglect to connect any cluster.

## 5 EXPERIMENTS

In our experiments we wish to answer questions regarding (1) the approximation quality of our model, (2) the quality of our automatic parameter selection, (3) the benefit from learning versus A-NN for large scale applications and (4) the scalability of the model. As this contribution is the first to scale a graph learning algorithm to a large amount of nodes, it is difficult to compare with other methods without trying to scale them first. While this paper focuses on the scaled log model, we also scaled the $\ell_2$-model and the "hard" and "soft" models proposed by (Daitch et al., 2009). We refer the reader to the supplementary material C for further details, and D for details about the datasets we use.

### 5.1 APPROXIMATION QUALITY OF LARGE SCALE MODEL

When computing $\mathcal{E}^{\text{allowed}}$, we use an A-NN graph from the publicly available FLANN library[1] that implements the work of Muja & Lowe (2014). To learn a graph with on average $k$ neighbors per node ($k$-NN), we first compute an $rk$-A-NN graph and use its edges as $\mathcal{E}^{\text{allowed}}$. The graph is then learned on this subset. The size of $\mathcal{E}^{\text{allowed}}$ does not only affect the time needed to learn the graph, but also its quality. A too restrictive choice might prevent the final graph from learning useful edges. In Figure 3, we study the effect of this restriction on the final result. The vertical axis is the relative $\ell_1$ error between our approximate log

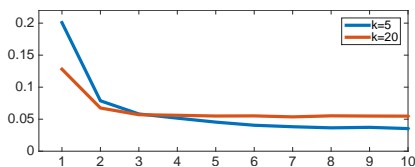

Figure 3: Approximation error between our large scale log model and the exact log model by Kalofolias (2016).

model and the actual log model by Kalofolias (2016) when learning a graph between 1000 images of MNIST, averaged over 10 runs. Note that the result will depend on the A-NN algorithm used, while a comparison between different types of A-NN is beyond the scope of this paper.

### 5.2 EFFECTIVENESS OF AUTOMATIC PARAMETER SELECTION

As we saw in the middle and right plots of Figure 2, the approximate bounds of $\theta$ (red line) given by eq. (17) are very effective at predicting sparsity. The same experiment is repeated for more datasets is in the supplementary material. In Figure 4 we see this also in *large scale*, between 262K nodes from our "spherical" dataset. Please note, that in the rare cases that the actual sparsity is outside the predicted bounds, we already have a good starting point for finding a good $\theta$. Note also that small fluctuations in the density are tolerated, for example in $k$-NN or A-NN graphs we always obtain results with slightly more than $nk$ edges due to the fact that $W$ is symmetric. Finally, as we see in Figure 11, the bounds are very *robust to outliers, as well as duplicates* in the data. The curious reader can find details in Section D.3.1 of the supplementary material.

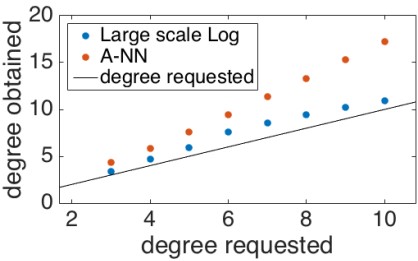

Figure 4: Effectiveness of $\theta$ bounds eq. (17). Requested versus obtained degree, "spherical" data ($262,000$ nodes).

### 5.3 EDGE QUALITY

We first compare scalable models of graphs between $60,000$ images of MNIST. MNIST has a relatively uniform sampling between numbers of different labels, **except for the digits "1" that are**

---

[1]Compiled C code run through Matlab, available from `http://www.cs.ubc.ca/research/flann/`.

**more densely sampled**. That is, the average intra-class distance between digits "1" is smaller than for other digits (supplementary material D.2). As we see, this affects the results of graph learning.

In Figure 5, we plot the histograms of connectivity between images of each label. We normalize all graphs to $\|W\|_{1,1} = 1$. The last bar is the connectivity wasted on wrong edges (pairs of different labels). The A-NN graph does not take into account the different sampling rates of different digits, while it has many wrong edges. The $\ell_2$ model (4) assigns most of its connectivity to the label "1" and neglects digits with larger distance ("2", "8") even for 30 edges per node. The log model does not suffer from this problem even for sparse graphs of degree 5 and gives consistent connectivities for all digits. The scaled versions of models in (Daitch et al., 2009) perform worse than our large scale log model, but often better than A-NN or $\ell$-2, as plotted in Figure 13 of the supplementary material.

Figure 6 (left) summarizes the number of wrong edges with respect to $k$. Models $\ell_2$ and log that minimize $\mathrm{tr}(X^\top LX) = \|Z \circ W\|_{1,1}$ have a large advantage compared to models by Daitch et al. (2009) that minimize $\|LX\|_F^2$ a quadratic function of $W$. The first induces sparsity of $W$, while the latter favors small edges. This explains the existence of many wrong edges for models "hard" and "soft", while creates problems in controlling sparsity. Note that A-NN is more accurate than "hard" and "soft" graphs. It seems that $\ell_2$ is more accurate than the log model, but as shown in Figure 5, the majority of its edges connect digit "1" without connecting sufficiently digits such as $2, 3, 8$.

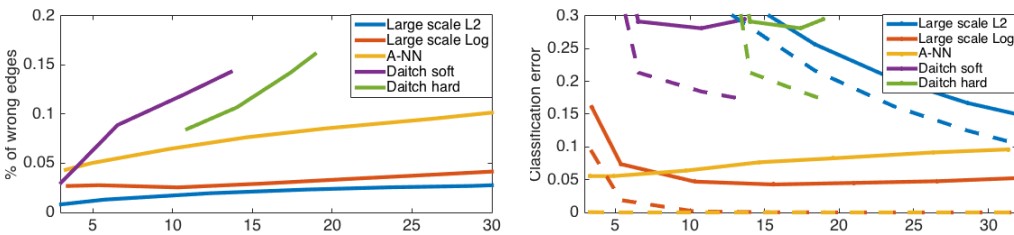

Figure 6: **Left:** Edge accuracy of large scale models for MNIST. **Right:** Digit classification error with $1\%$ labels. Dashed lines represent nodes in components without known labels (non-classifiable).

### 5.4 SEMI-SUPERVISED LEARNING

On the same graphs, we perform label propagation (Zhu et al., 2003) on MNIST, with $1\%$ known labels. The results are plotted in Figure 6 (right). Note that in label propagation, knowledge is not shared between disconnected components, making it impossible to classify nodes in components without known labels. The dashed lines of Figure 6 represent the number of nodes in such components, that occur much less in the log graph. Again, the log model performs best. To have a fair comparison with A-NN, we weighted its edges with standard exponential decay $W_{i,j} = \exp(-Z_{i,j}/\sigma^2)$ and chose every time the best performing $\sigma$. Note that it is very difficult for the "soft" and "hard" models to control sparsity (hard graphs have no parameter at all). Hence they are less adaptable to a particular problem and in this particular case perform poorly.

### 5.5 MANIFOLD RECOVERY QUALITY

Graphs can be used to recover a low dimensional manifold from data. We evaluate the performance of our large scale model for this application on three datasets: "*spherical data*" ($n = 262, 144$) and "*spherical data small*" ($n = 4, 096$) from a known 2-D manifold, and "*word2vec*" ($n = 10, 000$).

### 5.5.1 LARGE SPHERICAL DATA

Using 1920 signals from a known spherical manifold (Perraudin et al., 2018) we learn graphs of $262, 144$ nodes and recover a 2D manifold using the first 2 non-trivial eigenvectors of their Laplacians. The data is sampled on a $512 \times 512$ grid on the sphere, so we use graphs of degree close to $k = 4$. We try to recover it using scalable models: $\ell_2$ ($k = 4.70$), log ($k = 4.73$) and A-NN ($k = 4.31$).

While the A-NN graph performs very poorly, the representations recovered by both $\ell_2$ and log graphs are almost perfectly square (Figure 15). However, as shown in Figure 7, the log graph gives the best representation. We plot subgraphs of the two models with only the nodes that correspond to 2D grids in the middle or the corner *of the the original manifold*. While the $\ell_2$ graph does not connect all central nodes (green), the log graph is closer to a 2D grid. Furthermore, while the $\ell_2$ model had 46 disconnected nodes that we discarded before computing eigenvectors, the log graph was connected.

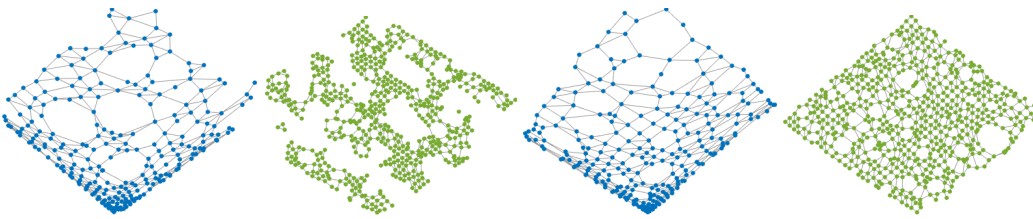

Figure 7: Detail from the manifolds recovered by $\ell_2$ and log models from "spherical data" ($262, 144$ nodes, 1920 signals). Corner (**blue**) and middle (**green**) parts of the manifold. **Left**: $\ell_2$ model, $k = 4.70$. **Right**: log model, $k = 4.73$. See Figure 15 for the big picture.

### 5.5.2 SMALL SPHERICAL DATA

We then learn graphs from the "small spherical data" so as to be able to compute graph diameters. This operation has a complexity above $O(n^2)$ and is not possible for large graphs. Manifold-like graphs are known to have larger diameter compared to small world graphs (Watts & Strogatz, 1998) for the same average degree.

In Figure 8 (left and middle) we plot the diameter of scalable graph models. The data has 4096 nodes organized as a $64 \times 64$ grid, therefore a ground-truth 4-NN has exactly diameter $127 = 2 \cdot 64 - 1$, and a ground-truth 8-NN diameter 63. The learned graphs, given enough information (1920 signals) reach the ideal diameter around degrees 4 or 8, with a phase transition from diameter 127 to diameter 63 between degrees 6 to 7. When the information is not enough to recover the exact manifold (using only 40 signals, middle plot of Figure 8), the learned graphs have a higher diameter than the A-NN graph.

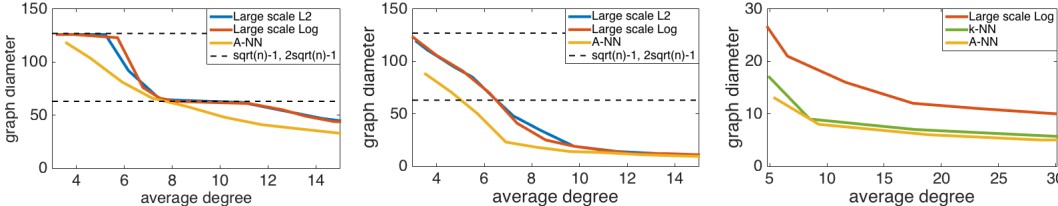

Figure 8: Graph diameter measures manifold recovery quality. **Left:** *small spherical data*: 4096 nodes, 1920 signals. **Middle:** Same data, 40 signals. **Right:** *word2vec*: 10,000 nodes, 300 features.

### 5.5.3 GRAPH BETWEEN WORDS USING WORD2VEC REPRESENTATIONS

Finally we learn a graph between $10,000$ words using 300 word2vec features, like the ones used for graph convolutional neural networks (Defferrard et al., 2016). In this application, connectivity of all nodes is important, and $\ell_2$ graphs failed, always giving many disconnected nodes.

In Figure 8 we compare the diameter of our large scale log graphs against A-NN and the $k$-NN graphs used by (Defferrard et al., 2016). The large scale log graph has significantly larger diameter than both $k$-NN and A-NN graphs. This indicates that our learned graph is closer to a manifold-like structure, unlike the other two types that are closer to a small-world graph. Manifold-like graphs reveal better the structure of data, while small world graphs are related to randomness (Watts & Strogatz, 1998).

In Figure 9, we plot nodes within two hops from the word "use" in the three types of graphs of degree around $k = 5$. We see that the NN graphs span a larger part of the entire graph just with 2 hops, the A-NN being closer to small-world. While the $k$-NN graph does better in terms of quality, it is significantly worse than our large scale log graph, that is actually cheaper to compute. Additionally, graph learning seems to assign more meaningful weights as we show in Table 1 of the supplementary material D.4.

### 5.6 COMPUTATION TIME

In Figure 1 we compare our scaled model to other graph learning models in terms of time as a function of $n$. For all iterative algorithms, except A-NN, we perform a maximum of 500 iterations. The scaled $\ell_2$ needs time per iteration identical to the scaled log model, but typically needs more

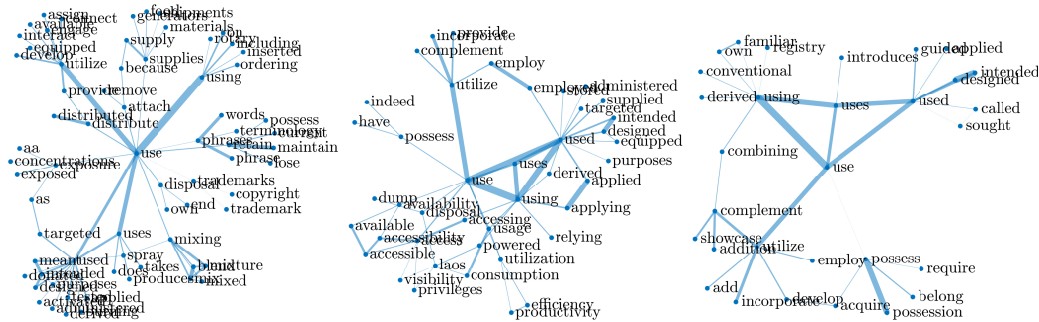

Figure 9: A 2-hop sub-graph of the word "*use*". **Left:** A-NN ($k = 5.4$). **Center:** $k$-NN graph ($k = 5.0$). **Right:** Large scale log ($k = 5.7$) being manifold-like only reaches relevant terms.

iterations (more than $5,000$ compared to $500$) till convergence. As we fix the number of iterations, they overlap and we only plot the log model.

In Figure 1 (left) we learn graphs between words, using $300$ features (*word2vec*). The cost is almost linear for our method, but quadratic for the original log model of Kalofolias (2016). The "hard" model corresponds to a close implementation of the original work of (Daitch et al., 2009). The most expensive part of the computation comes from the quadratic program solver. For the "soft" model, we used the forward backward scheme of (Beck & Teboulle, 2009) and the cost is governed by evaluating the KKT conditions to find the optimal support ($\mathcal{O}(n^2)$), that is faster than their original algorithm.

We also compare to our scaled versions of the "hard" and "soft" models, where we fix the support in the same way as explained in Section 3 (cf. supplementary material C). The bottleneck of the computation comes from the term $\|LX\|_F^2$ that requires $\mathcal{O}(knd)$ operations as opposed to our $\mathcal{O}(kn)$ for $\mathrm{tr}\left(X^\top LX\right)$. To reduce this cost, we randomly projected the original $d = 300$ input features to a subspace of size $d = 20$. While the cost per iteration of the "soft" method is slower than for the "hard" method, the latter typically needs more iterations to reach convergence.

In Figure1 (right), we show the scalability of our model to graphs of up to $1$**M samples**[2] of *US census*. Setting $k = 5$, it used **16 minutes** to perform $500$ iterations of graph learning on a desktop computer running Matlab. While this is a proof of scalability, for really large data one will have to consider implementation details like memory management, and using a faster programming language (like C++). Note that the A-NN implementation we used was compiled C code and thus much faster. In Figure 14 we illustrate the linear scalability of our model w.r.t. the size of the set of allowed edges.

## 6 CONCLUSIONS

We propose the first scalable solution to learn a weighted undirected graph from data, based on A-NN and the current state-of-the-art graph learning model. While it costs roughly as much as A-NN, it achieves quality very close to state-of-the-art. Its ability to scale is based on reducing the variables used for learning, while out automatica parameter selection eliminates the need for grid-search in order to achieve a sought graph sparsity. We assess its quality and scalability by providing an extensive set of experiments on many real datasets. The new large scale graphs perform best in various machine learning cases. They give better manifolds, are better for semi-supervised learning, and select the right amount of edges without allowing disconnected nodes. Learning a graph of 1 million nodes only takes 16 minutes using our simple Matlab implementation on a desktop computer.

### THANKS

The authors would like to especially thank Pierre Vandergheynst for his helpful comments during the preparation of this work at LTS2. We thank also the ETHZ Cosmology Research Group for allowing us to use their "Spherical convergence maps dataset" in the manifold recovery experiment.

---

[2]We provide code for our algorithm online in the GSPBox (Perraudin et al., 2014). A tutorial based on this paper can be found in `https://epfl-lts2.github.io/gspbox-html/doc/demos/gsp_demo_ learn_graph_large.html`.

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

# LARGE SCALE GRAPH LEARNING FROM SMOOTH SIGNALS: SUPPLEMENTARY MATERIAL

## A    PROOF OF THEOREM 1

**Proof.** *For any edge $i, j$ the optimality conditions are:*

$$2\theta Z_{i,j} + 2W_{i,j}^* - \frac{1}{\sum_{k \neq i} W_{i,k}^*} - \frac{1}{\sum_{k \neq j} W_{k,j}^*} = 0.$$

*Let us select any $i, j$ such that $W_{i,j} > 0$. Define $\eta_{i,j}$ so that*

$$2\eta_{i,j} = \frac{2}{W_{i,j}^*} - \frac{1}{\sum_{k \neq i} W_{i,k}^*} - \frac{1}{\sum_{k \neq j} W_{k,j}^*}, \tag{18}$$

*and note that $\eta_{i,j} \geq 0$, because all elements of $W^*$ are non-negative and $\frac{1}{W_{i,j}^*} \geq \frac{1}{W_{i,j}^* + \sum_{k \neq i} W_{i,k}^*}$.*
*Then we can rewrite the optimality conditions so that they can be solved analytically as a function of the unknown $\eta_{i,j}$:*

$$2\theta Z_{i,j} + 2W_{i,j}^* - \frac{2}{W_{i,j}^*} + 2\eta_{i,j} = 0 \Rightarrow$$

$$W_{i,j}^* = \frac{\sqrt{(\theta Z_{i,j} + \eta_{i,j})^2 + 4} - (\theta Z_{i,j} + \eta_{i,j})}{2} \leq 1, \tag{19}$$

*since $\sqrt{4 + x^2} - x \leq 2, \forall x \geq 0$. This inequality holds for any edge $i, j$ of the learned graph such that $W_{i,j}^* > 0$, therefore we can write*

$$\max_{i,j} W_{i,j}^*(\theta Z, 1, 1) \leq 1. \tag{20}$$

*Hence using proposition 1, we conclude the proof.*

Note that for sparsely connected regions, $\eta_{i,j}$ becomes smaller, and therefore the edges of the graph come closer to the upper limit. We can deduct from equation 19 that the limit is actually reached only in case of duplicate nodes, i.e when $Z_{i,j} = 0$, and if the rest of the nodes are sufficiently far so that the edge $i, j$ is the only edge for both nodes (so that $\eta_{i,j} = 0$).

## B    PROOF OF THEOREM 3

**Proof.** *From the proof of Theorem 2, we know that $\|w^*\|_0 = k$ if and only if $\lambda^* \in [\theta z_k, \theta z_{k+1})$. We can rewrite this condition as*

$$\theta z_k \leq \frac{\theta b_k + \sqrt{\theta^2 b_k^2 + 4k}}{2k} < \theta z_{k+1} \Leftrightarrow$$

$$2k\theta z_k - \theta b_k \leq \sqrt{\theta^2 b_k^2 + 4k} < 2k\theta z_{k+1} - \theta b_k \Leftrightarrow$$

$$4k^2\theta^2 z_k^2 - 4k\theta^2 b_k z_k \leq 4k < 4k^2\theta^2 z_{k+1}^2 - 4k\theta^2 b_k z_{k+1} \Leftrightarrow$$

$$\theta^2(kz_k^2 - b_k z_k) \leq 1 < \theta^2(kz_{k+1}^2 - b_k z_{k+1}).$$

*As $\theta$ is constrained to be positive, the only values that satisfy the above inequalities are the ones proposed in the theorem.*

## C    SCALING ALGORITHM OF (DAITCH ET AL., 2009)

(Daitch et al., 2009) proposes two convex optimization problems in order to learn graph from smooth signals. The prior used is $\|LX\|_F^2$, which is a quadratic function of the weight $W$. For the first problem, corresponding to the "hard" model, each node is constraint to have a degree of at least 1:

$$\underset{w}{\text{minimize}} \|Mw\|_2^2 \qquad \text{s.t. } w \geq \mathbf{0}, Sw \geq \mathbf{1}, \tag{21}$$

where $M$ is a linear operator such that $\|LX\|_F^2 = \|Mw\|_2^2$. We refer to the original paper for the construction of $M$. The second problem, corresponding to the "soft" problem, is a relaxed version of the "hard" one where each every degree below 1 is penalized quadratically. It reads:

$$\underset{w}{\text{minimize}} \|Mw\|_2^2 + \mu \| \max(\mathbf{1} - Sw, 0)\|_2^2 \qquad \text{s.t. } w \geq \mathbf{0}, \tag{22}$$

where $\mu$ is a parameter weighting the importance of the constraint. In comparizon to the log and the $\ell_2$ models where changing the regularization parameters has an important effect on the final average node degree, we note that $\mu$ has little effect on the final average node degree. In their original paper, Daitch etal solve their optimization problems using SDPT3 (Tütüncü et al., 2003), which has a complexity significantly higher than $\mathcal{O}\left(v^2\right)$ for $v$ variables. So even when the number of edges scales with the number of nodes, i.e. $v = \mathcal{O}\left(n\right)$, the complexity is not better than $\mathcal{O}\left(n^2\right)$. To minimize this issue, one of the important contribution of (Daitch et al., 2009) is to solve the problem on subset $v$ of edges. Nevertheless, instead of keeping the support fixed (as we do), they propose to check the KKT conditions of the dual variable, they can assess if the some edges should be added to the support at end optimization scheme. If so, the optimization is run again with the new edges. The process is repeated until all KKT conditions are satisfied. This technique allows for finding the global solution of larger problems, (particularly if SDPT3 is used). We note however that the search of the next relevant support costs $O(n^2)$ again, since they have to compute the KKT conditions for all possible edges.

We started by implementing the original "hard" and "soft" algorithms and obtained similar speed that were comparable as the results reported in the original paper. Unfortunately, we were not able to run theses implementations for more than a few thousands of nodes. Hence, in order to compare with our algorithm, we had to cope with their scalability issues first. Essentially, we derived new algorithms thanks to two modifications. First, we used FISTA ((Beck & Teboulle, 2009)) and forward-backward-based primal-dual (Komodakis & Pesquet, 2015, Algorithm 2) optimization schemes respectively for the soft and hard graph optimization instead of SDPT3. Second, we removed the support optimization using the KKT conditions and use the same A-NN support as our algorithm. After those modification, our implementation scaled to approximately a $100'000$ nodes. While the theoretical complexity is about the same as for our optimization problem, the running time are significantly longer than our method because the term $\|LX\|_F^2 = \|Mw\|_2^2$ requires $d$ times more computation than $\text{tr}\left(X^T LX\right) = \|Zw\|_1$, where $d$ is the number of signals. In order to be able to compare to these algorithm in Figure 1, we randomly projected the original $d = 300$ input features to a subspace of size $d = 20$ (this does not change significantly the computation time of our model). While the cost per iteration of the "soft" method is slower than for the "hard" method, the latter typically needs more iterations to reach convergence. The reason is that our accelerated algorithm uses FISTA that has a global rate of convergence proven to be significantly better than the forward-backward based primal-dual we used (Beck & Teboulle, 2009; Komodakis & Pesquet, 2015).

## D    EXPERIMENTS

### D.1    DATASETS

- **MNIST:** We use the 60000 images of the MNIST training dataset.
- **US Census 1990:** Dataset available at the UCI machine learning repository, consists of approximately 2.5 million samples of 68 features. `https://archive.ics.uci.edu/ml/datasets/US+Census+Data+(1990)`
- **word2vec:** The $10'000$ most used words in English (`https://research.googleblog.com/2006/08/all-our-n-gram-are-belong-to-you.html`). It uses the Google word2vec features (`https://code.google.com/archive/p/word2vec/`).
- **Spherical data:** Simulated cosmological mass maps (i.e. the amount of mass in the universe observed from earth in every direction) from Perraudin et al. (2018); Sgier et al. (2018). The dataset consists of two sets of maps that were created using the standard cosmological model with two sets of cosmological parameters. This data resides on the sphere, which can be considered as its underlying manifold. We used the slightly smoothed maps (Gaussian kernel of radius 3 arcmin), as proposed in the original paper. While the original data consists of 40 maps of size $12 \cdot 1024^2$, we work only a sup-part of the sphere of size. This allow us

to build 1920 signals on a sub-part of the sphere ($512 \times 512$ grid), i.e. between $262, 144$ nodes. The actual manifold with the correct coordinates is plotted in Figure 15 up left. `https://zenodo.org/record/1303272`

- **Small spherical data:** A subset of "Spherical data" containing 4096 nodes, (64x64 grid) and the same 1920 signals.
- **USPS:** Handwritten digits from 0 to 9. `https://ieeexplore.ieee.org/document/291440`

## D.2 MNIST IRREGULAR INTRA-CLASS DISTANCES

Figure 10 illustrates one irregularity of the MNIST dataset. One could expect that the average distance between two digits of the same class (intra-class) is more or less independent of the class. Nevertheless, for the MNIST dataset, the distance between the 1 is significantly smaller than the one of the other digits. For this reason, the L2 model connects significantly more the digits 1 than the others.

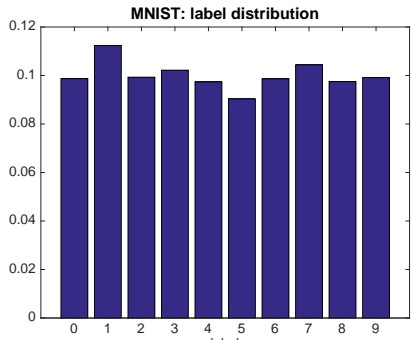 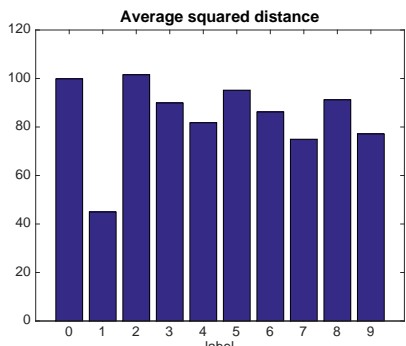

Figure 10: Label frequency (**left**) and average squared distribution (**right**) of MNIST train data (60000 nodes). The distances between digits "1" are significantly smaller than distances between other digits.

## D.3 APPROXIMATION ACCURACY AND ROBUSTNESS OF PARAMETER $\theta$

We already saw in Figure 2 (middle) that for the MNIST dataset (17) predicts very well the sparsity of the final graph for any choice of $\theta$. This is further illustrated on the USPS and ATT faces datasets in Figure 12. Note, that in the rare cases that the actual sparsity is outside the predicted bounds, we already have a good starting point for finding a good $\theta$. For example, in the COIL dataset, if we want a graph with 15 edges per node we will set $\theta = 1.2$, obtaining instead a graph with 12 edges per node. This kind of fluctuations are usually tolerated, while even in $k$-NN we always obtain graphs with more than $nk$ edges due to the fact that $W$ is symmetric.

### D.3.1 ROBUSTNESS TO OUTLIERS AND DUPLICATES

We test the robustness of the bounds of $\theta$ by repeating the experiment of Figure 2 (middle) after replacing $10\%$ of the images with outliers or duplicates. For this experiment, we first added outliers by contaminating $10\%$ of the images with Gaussian noise from $\mathcal{N}(\iota, \infty)$. Note that this is huge noise since the initial images have intensities in the range $[0, 1]$. The result, plotted in Figure 11 (left), is almost identical to the results without outliers (Figure 2, middle).

While this might seem surprising, note that $\theta$ given by eq. (17) are really dependent on the smallest distances in $Z$, rather than the largest ones: $\hat{Z}$ is sorted, and B as well. Adding outliers induces additional nodes distances larger than usually, that never make it to change the first $k$ rows of matrices $\hat{Z}$ and $B$, for the columns that correspond to non-outlier nodes. *Therefore, eq. (17) is very robust to outliers*.

To complete the experiment, instead of adding noise, we replaced $10\%$ of the images with copies of other images already in the dataset. In this case, we have $20\%$ of images in *pairs of duplicates*, with essentially zero distances with each other in matrix $Z$. As we see in Figure 11 (right), while the intervals of $\theta$ change, they do so *following closely the actual measured sparsity*.

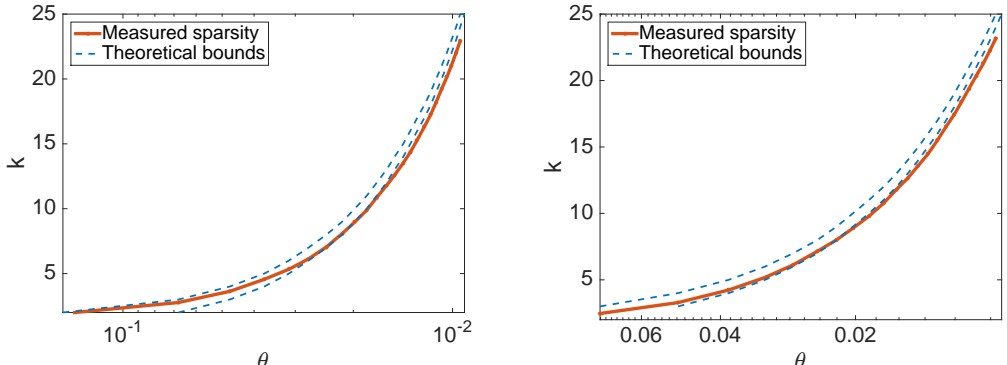

Figure 11: Robustness of the theoretical bounds of $\theta$ in the existence of outliers or duplicate nodes. Same dataset as the one used for Figure 2. Even for extreme cases in terms of distance distribution, the bounds give a good approximation. **Left**: Results when we add Gaussian noise from $\mathcal{N}(0, 1)$ to 10% of the images before calculating $Z$. Note that the noise added is significant given that the initial pixel values are in $[0, 1]$. **Right**: We replaced 10% of the images with duplicates of other images already in the dataset.

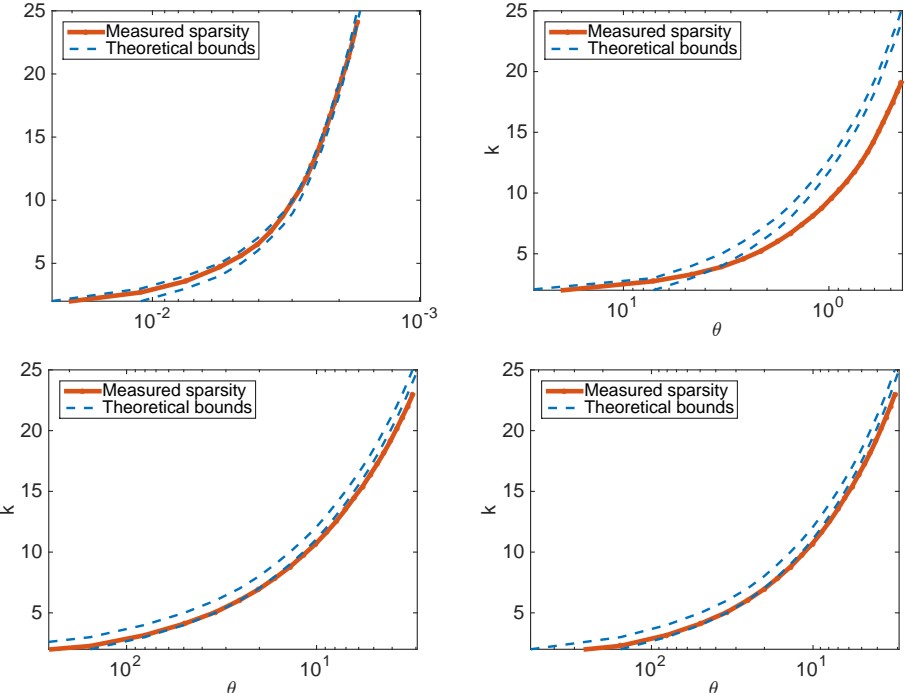

Figure 12: Predicted and measured sparsity for different choices of $\theta$. Note that $\theta$ is plotted in logarithmic scale and decreasing. **Up left**: 400 ATT face images. **Up right**: 1440 object images from the COIL dataset. **Down left**: Graph between 1000 samples from a multivariate uniform distribution. **Down right**: Graph between 1000 samples from a multivariate Gaussian distribution.

### D.4 CONNECTIVITY EXAMPLE OF THE GRAPH OF WORDS

In Table 1, we look in more detail at the graph constructed from the word2vec features. We present the connectivity for the word "glucose" and "academy". Looking at different words, we observe that the learned graph is able to associate meaningful edge weights to the different words according to the confidence of their similarity.

| Word | $k$-NN | A-NN | Learned |
|------|--------|------|---------|
| glucose | 0.1226 insulin
0.0233 protein
0.0210 oxygen
0.0148 hormone | 0.0800 insulin
0.0337 protein
0.0306 oxygen
0.0295 cholesterol
0.0263 calcium
0.0225 hormone | 0.5742 insulin
0.0395 calcium
0.0151 metabolism
0.0131 cholesterol |
| academy | 0.0996 training
0.0953 school
0.0918 institute | 0.0901 young
0.0863 department
0.0841 bizrate | 0.3549 training
0.2323 institute
0.1329 school
0.0135 camp
0.0008 vocational |

Table 1: Weight comparison between $k$-NN, A-NN and learned graphs. The weights assigned by graph learning correspond much better to the relevance of the terms.

### D.5 MNIST CONNECTIVITY FOR (DAITCH ET AL., 2009) METHODS

In Figure 13, we plot the connectivity across different digits of MNIST for the "hard" and "soft" model. As the degree is constant over the nodes, the hard model perform similarly to the A-NN (see Figure 5). It seems that due to hard constraint, the "hard" model forces many edges with a small weights. On the other hand, in terms of connextivity, the soft model seems to be between the log and the $\ell_2$ model.

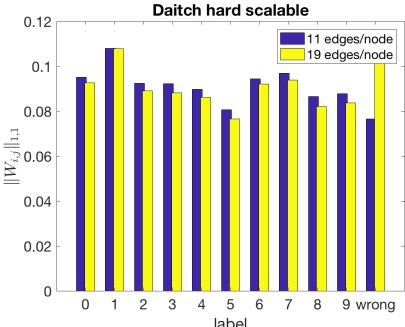 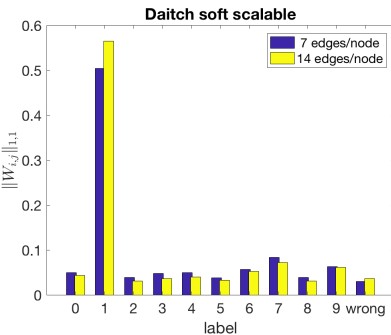

Figure 13: Connectivity across different classes of MNIST (60000 nodes). The graph is normalized so that $\|W\|_{1,1} = 1$. We measure the percentage of the total weight for connected pairs of each label. The last columns correspond to the total of the wrong edges, between images of different labels. **Left**: (Daitch et al., 2009) hard model. As the degree is constant over the nodes, the hard model is close the A-NN. **Right**: (Daitch et al., 2009) soft model. In terms of connextivity, the soft model seems to be between the log and the $\ell_2$ model. Note that while it favors connections between "1"s, this effect becomes worse with higher density. Note also that these algorithms fail to give reasonable graphs for densities outside a small range, making it very difficult to control sparsity.

### D.6 MNIST COMPUTATIONAL TIME WITH RESPECT TO K

The cost of learning a graph with a subset of allowed edges $\mathcal{E}^{\text{allowed}}$ is linear to the size of the set as illustrated in Figure 14. For this experiment, we use the MNIST data set. To learn a graph with approximately 10 edges per node, we needed 20 seconds to compute $\mathcal{E}^{\text{allowed}}$, and 20 seconds to learn the final graph of 60000 nodes (around 250 iterations). Note that the time necessary to search for the nearest neighbors is in the same order of magnitude than the learning process.

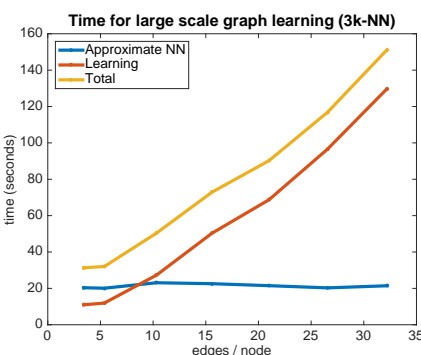

Figure 14: Time needed for learning a graph of 60000 nodes (MNIST images) using the large-scale version of (3). Our algorithm converged after 250 to 450 iterations with a tolerance of $1e-4$. The time needed is linear to the number of variables, that is linear to the average degree of the graph.

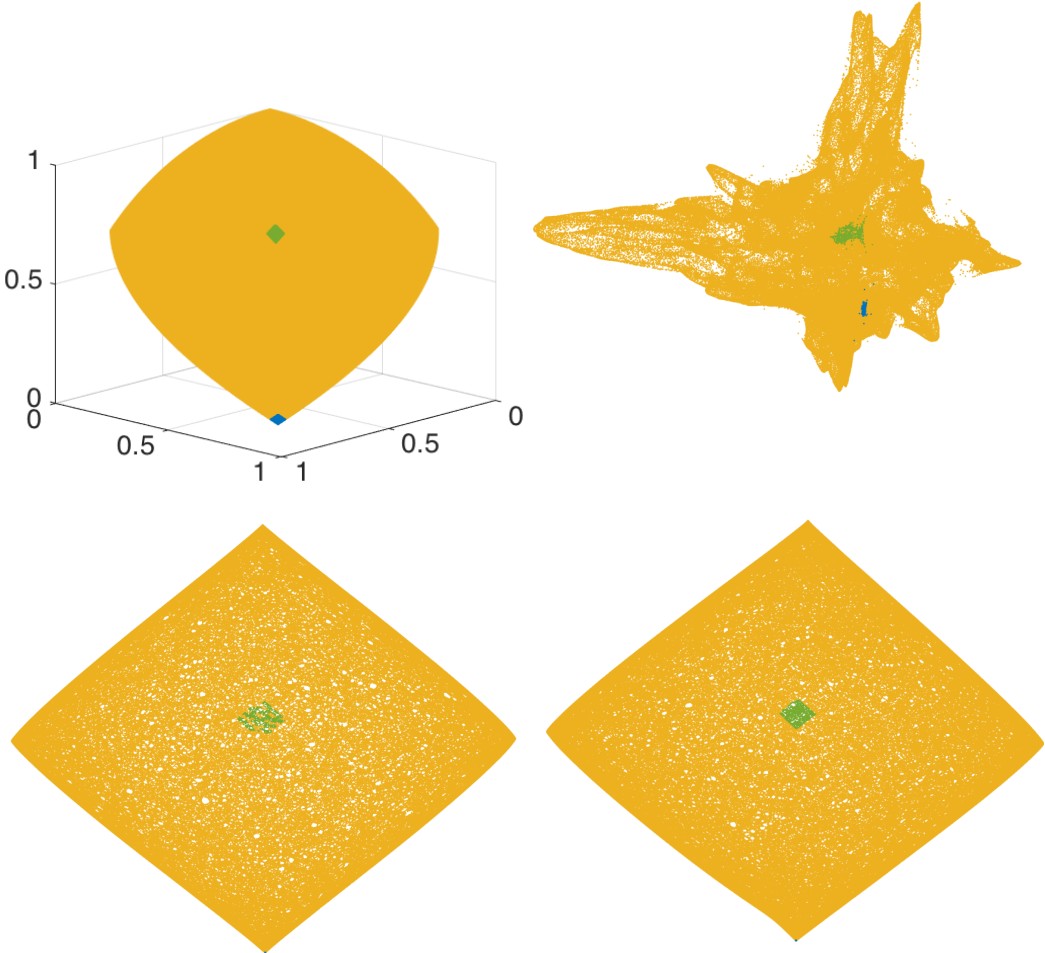

Figure 15: Spherical data, ground truth and recovered manifolds. **Up left:** The ground truth manifold is on the sphere. We have colored the nodes that correspond to the middle of the 2-D grid and the lower corner so that we track where they are mapped in the recovered manifolds. In Figure 7 we keep only the subgraphs of the green or blue nodes. **Up, right:** Recovered by A-NN, $k = 4.31$. **Down, left:** Recovered by the $\ell_2$ model, $k = 4.70$. The middle region is mixed with nodes outside the very center. The corners are much more dense, the blue region is barely visible on the bottom. Note that 46 nodes were disconnected so they are not mapped at all. **Down, right:** Recovered by the log model, $k = 4.73$. The middle region is much better mapped. The corners are still very dense, we have to zoom-in for the blue region (Figure 7).

