# OpenReview forum: "Large Scale Graph Learning From Smooth Signals"
_ICLR.cc/2019/Conference_

### Official Review · AnonReviewer2 · 2018-11-02
**Good paper, very well written**

**Rating:** 7
**Confidence:** 4

**Review:**

This paper proposed an approximation technique to learn the large-scale graph with the desired edge density. It was well-written and contains thorough experimental results and analysis.

A minor drawback is that while this work was motivated by the use of k-NN graph in graph convolution network (GCN), there was no evidence on how well A-NN performs in compare to k-NN with GCN.

---

> ### Author Response · Authors · 2018-11-19
> **Answer to reviewer**
>
> Dear reviewer,
> Thank you for your review and your positive comments.
>
> Comparing the performance of GCN using different graphs (k-nn, A-NN, learned) is definitely an interesting and relevant topic. While going deep to GCNs is out of the scope of this paper, we took a step in this direction by learning a graph that could be used for deep learning.
>
> In the new version of the paper we are adding a new experiment, where we work with the data used by [Perraudin etal 2018] with the DeepSphere GCN. Because the graph used in DeepSphere is constructed using empirical rules and is not a real ground truth, one could ask themselves, if there exists a better way to construct it. The graph we learned using our algorithm would be a candidate. We show in our experiments that the actual graph that the authors used for deep learning (that they constructed knowing the coordinates) is close to the one we learned from smooth signals, without having any information of coordinates. Furthermore, the sphere is a 2D manifold, and our graph has properties similar to a 2D manifold graph.

---

### Official Review · AnonReviewer3 · 2018-11-02
**marginally bellow threshold**

**Rating:** 5
**Confidence:** 3

**Review:**

The paper proposes a scalable approximate calculation of graph construction. Based on the sparse optimization formulation of a graph construction, the authors provide a way to select parameter automatically based on user desired connectivity of graph.

The problem setting, graph construction, is significant for the wide range of ML community. Overall, however, advantage/novelty of the proposed method is unclear for me.

Scalability is main advantage of the proposed method, but the authors just employed known nearest neighbor approximation methods, and thus here no technical novelty is shown.

I couldn't find connection between Section 3 and 4, these seem to be an independent topics. Main claim of the paper would be in Section 3, but the novelty would be weak as mentioned above.

Solving reverse problem is interesting, but it just provide the parameter value range which results in given sparsity level k. This doesn't provide exact value of \theta (and user still have to specify k), and selection would be possible easily without the analytical formula (e.g., by following the regularization path)

Performance verification is not convincing. Showing accuracy gain for more wide variety of datasets would be convincing.

---

> ### Author Response · Authors · 2018-11-19
> **Answer to reviewer (Part 1)**
>
> Dear reviewer,
> We would like to first of all thank you for your review. We completely agree that graph construction is a significant problem for the wide range of ML community. From your comments we understand that we have to make more obvious for the reader the novelty of our paper and its importance for the ML community.
>
> Novelty/contribution:
> --------------------------
> Until now, there was no real graph learning algorithm in the literature that actually scales to problems larger than a few thousands of nodes. The previous state of the art algorithms, but also classic algorithms like the ones of Daitch etal (see comments to Reviewer 1) cannot scale due to their computational complexity that is in the order at least O(n^2). For our experiments with the MNIST dataset (60,000 nodes) none of them could be computed. In the new version of the paper, we were able to compare with Daitch only because we modified significantly their algorithms.
>
> Alternative to graph learning, people could resort to approximate nearest neighbor (A-NN) algorithms, that do not weight the edges, but only return a binary approximate adjacency matrix. These algorithms are more practical for large scale problems, but suffer in quality of the edges.
>
> Our solution is the first actual graph learning solution (based on tr(X^TLX)), that thanks to A-NN scales with O(dn logn) for n nodes and d features. Due to this complexity it can scale to learn graphs of 1 million nodes in reasonable time on a laptop running Matlab. To the best of our knowledge, there is no other graph learning algorithm that could scale to this size of graphs.
>
> To achieve this scalability, our contribution is twofold (first, Section 3 and second, Section 4).
>
> First, in Section 3, we show how the optimization problem of the previous state of the art can be reduced if we know a reduced support of edges. While this is small part of our contribution, we show the details of the optimization, analyze the new computational complexity, and provide experiments to show the quality of the approximate graphs in realistic scenarios one would come across in ML. Furthermore, we will provide online our Matlab code so that the ML community can learn large scale graphs for their own data.
>
> Secondly, in Section 4, we reduce a big drawback of the previous state of the art model. The latter suffers from the problem of how to set the two parameters $\alpha$ and $\beta$. This is a real problem when we don't even know the order of magnitude of the parameters, and the only solution seems to be trial and error, a.k.a. grid search over both parameters, in a logarithmic scale. In real ML scenarios, we want to be able to set the density of the graph, for example to 5 or 10 nearest neighbors in average. Having to try 25 or 100 different settings (5 or 10 for $\alpha$ times 5 or 10 for $\beta$ for [Kalofolias 2016]) would be a prohibitive factor for large scale problems.
>
> Even when these two parameters are reduced to the more intuitive ones $\theta, \delta$ (Proposition 1 in our paper), there was no way to know the order of magnitude of $\theta$ for controlling sparsity. In our experiments we had to use values of theta in the order of magnitude of $1e-6$ (Figure 2), but also of $1e2$ (Figure 10). As you propose, following the regularization path was the only way available until now, but this needs to run the algorithm many more times (like [Kalofolias, 2016] did in his paper). Hence our contribution is to propose a natural way to set theta in order to control the sparsity. We basically propose a method to link $theta$ and $k$, the number of desired edges per node.  While one may argue that $k$ still needs to be tuned, this parameter can be interpreted and is way simpler to set based on data assumptions than is impossible with $\theta$.
>
> Parameter value range:
> -------------------------------
> As you say, we give a range of parameters that shall give approximately a requested graph sparsity. In our experiments, we always use the geometric mean between the upper and lower bounds for each given k. This is equivalent to using the arithmetic mean in the log scale, that represents the order of magnitude of $\theta$, and is what we are plotting in Figure 2 and Figure 10 (logarithmic $\theta$ scale).
>
> We are adding this information in the main text, as you pointed out it is important to make it clear, and we thank you for this comment.
>
> [Continues in Part 2]

---

> ### Author Response · Authors · 2018-11-19
> **Answer to reviewer (Part 2)**
>
>
> Performance verification for more datasets:
> ---------------------------------------------------------
> As we wrote in the answer to reviewer 1, it is difficult to assess the quality of graphs when the ground truth is not really known. To assess quality, we tried to use a wide variety of both datasets and proxies for graph quality. We showed the distribution quality of edges for MNIST (Figure 3), the edge accuracy for MNIST (Figure 4) and the classification error of label propagation on MNIST (Figure 5). In the latter we also compared the number of disconnected nodes. We measured the diameter of the graph between word2vec representations (Figure 6) that is best for our large scale graph, and showed qualitatively the effects of the graph in Figure 7.
>
> Following your comment regarding performance verification, we added a further experiment, for larger data this time (262,000 nodes). We used our learned graph for embedding all nodes on a 2-D plane. The signals were known to reside on a small part of the surface of the sphere, which is a 2-dimensional manifold. Without giving any notion of coordinates, but only smooth signals of the sphere, we were able to recover a very good 2-D embedding by using the first two non-zero eigenvectors of our learned sparse Laplacian. This is an important result, since we used no coordinate information whatsoever, only the similarity between different nodes, and the structure was very well recovered.
>
> In this experiment it is clear that the large scale Log model works best. The large scale L-2 graph is able to recover a meaningful 2D manifold only if we remove the disconnected nodes, and in that case, it gives erroneous results in the middle of the manifold. The A-NN has no disconnected nodes, but gives an embedding that is far from 2D, as many of the edges are erroneous.
>
> In their work [Perraudin etal 2018] used in a weighted 8 exact K-NN graph for their experiments (Figure B.13 in their paper). Knowing this graph, we can see we can compare different algorithms with respect to the f1 measure. Again, we observed that the learned graphs perform significantly better than A-NN.
>
> We are adding the plots of the different embeddings, as well as the f1 scores in the body or in the appendix of our paper depending on the available space. Furthermore, we are adding a plot of expected versus obtained degrees of the graph using our theta approximations.

---

### Official Review · AnonReviewer1 · 2018-11-03
**Neat contribution**

**Rating:** 7
**Confidence:** 5

**Review:**

Learning graphs from data fine tunes standard similarity graph constructions such as k-nearest neighbor graphs.
There has been a line of research works that focuses on learning graphs and that shows that this results in superior
results in various machine learning tasks.  The current state-of-the-art method is the method proposed by Kalofolias,
which however is slow.   The authors suggest a method to avoid searching for the parameters that achieve a desired
level of sparsity by providing closed a formula. The parameter that determines the sparsity is theta, see proposition 1 on page 4. This was originally shown by Kalofolias. To achieve their goal, the authors first consider the degree of any given node by looking at equation (8), page 4. They prove theorem 1, that is intuitive and  provides the form of the optimal
weights that connect this node to the rest of the nodes in the graph.  The proof is based on applying the KKT conditions on
the objective (8), with the single constraint that there are no negative weights.  Finally, since we care about the
sparsity of the graph as a whole, the authors use the average of the parameter theta over all nodes. The authors perform
experiments on real-world graphs, and show basic properties of their method, as well as the main source of mistakes ,i.e., disconnected nodes, figure 5.

Essentially, this paper starts from the work of Kalofolias  and improves it significantly. This by itself is
a neat contribution, but the authors could improve their paper by showing a more complete view  of graph
learning methods, with respect to the quality of the produced graphs and the scalability. I find this aspect of the paper narrowing its contribution, hence my evaluation. Some remarks follow.

- A different family of graph learning methods is based on the objective ||LX||_F^2 or equivalently tr(X^TLLX).
For this objective, Daitch et al. proved certain neat properties, such as the existence of a sparse optimal graph.
This allows Daitch et al. to solve the primal dual significantly faster than O(n^2) since by their theorem,
O(nd) edges are required where d is the dimension of the data points. When d is large, a random projection can be applied.
The paper should compare with this family of methods that are more scalable both with respect to the accuracy,
and to the runtimes.

- While the proposed method scales significantly better than Kalofolias, the datasets used are small.

- Using LSH for k-nn graphs results in a  scalable, practical way to construct similarity graphs. The authors should cite
the following related work, and compare with such methods.
“Efficient K-Nearest Neighbor Graph Construction for Generic Similarity Measures“ by Dong, Charikar, Li.

- An interesting experiment would be to inject outliers in the dataset, or use some dataset with outliers.
Would this affect the tightness of the interval in equation (17)?

---

> ### Author Response · Authors · 2018-11-19
> **Answer to reviewer (Part 1)**
>
> Dear reviewer,
> We really appreciate your thorough review and the positive comments, as well as your propositions for improvements. We tried to address the points that you made in your review, and we believe that the paper is, as you suggested, stronger with these additions.
>
> ||LX||_F^2:
> ----------------
> The models of Daitch etal are indeed very important ones in the graph learning literature. Following your comments, we tried to compare both in accuracy and scalability. Note that there are actually two models in their paper, that they call “hard” and “soft” graphs. Hard graphs have the hard constraint that each node has degree at least 1. Soft graphs allow for degrees lower than 1, but penalize them quadratically.
>
> In their original paper, Daitch etal solve their optimization problem using SDPT3, which has a complexity significantly higher than O(v^2) for v variables. So even when v=o(n), the complexity is not better than O(n^2). Furthermore, in order to solve their problem in a more scalable way, Daitch etal restrict the support of the graph to a subset of edges. Then, by checking the KKT conditions on the dual variable, they can assess if the some edges should be added to the optimization scheme. If so, the optimization is run again. This is actually a very nice idea and it allows for solving larger problems, (particularly if SDPT3 is used). However, the search of the next relevant support costs O(n^2) again, since they have to compute the KKT conditions for all possible edges.
>
> We started by implementing the original hard and soft Daitch algorithm and obtained similar speed as reported in the original paper. Unfortunately, we were not able to run the algorithms for more than a few thousands of nodes. Hence, in order to cope with their scalability issues and provide some comparison with our models, we have derived a “more” scalable variant of Daitch algorithm. Essentially, we did two modifications. First, we removed the support optimization using the KKT conditions and use the same support as our algorithm.  Second, we used FISTA and primal dual optimization schemes respectively for the soft and hard graph optimization instead of SDPT3. This time, our implementation scaled to the order of a 100’000 nodes for a powerful desktop computer with 64Gb of RAM. The running times of optimization are still significantly higher than our models, because the term ||LX||_F^2 =||M w||_2^2 takes p times more computation than tr(X^T L X) = ||Z w||_1, where p is the number of signals.
>
> We can add the resulting time in Figure 1 of our paper: within 30 seconds (the new y limit of the plot), we were only able to learn a hard graph of 250 nodes using quadratic programming, and a soft graph of 2000 nodes when proximal splitting method were used. When the KKT conditions trick is not used, our version of Daitch algorithm were much faster but still significantly slower than our algorithm because of the dependencies on p. Note that we also used random projections to reduce the dimension from 300 to 20 only for the Daitch algorithms, while ours was running on the full set of 300 features
>
> In terms of quality, in our paper we focus on scalable algorithms (A-NN, scaled L2-degrees, scaled Log-degrees). Figures 4 and 5 of our paper show this comparison, that now we also run for the scaled version of the Daitch-soft model. In our experiments, we see that the Daitch model has many wrong edges in Figure 4, while in Figure 5 it performs slightly better than the L2 scaled model for specific edge densities, but always worse than the A-NN and the scaled Log-degrees model. Also, we see that Daitch suffers the same problem as the scaled L2 model: it has many disconnected nodes (less than L2, more than Log). This is expected, as the soft constraint is a quadratic one similar to the one of the L2 model, that allows node degrees to be zero.
>
> We believe that one major issue of the Daitch hard algorithm is that it does not provide a way to control sparsity. Hence we varied the size of the support to get different graphs. As for the soft algorithm, the regularization parameter controls the strength of the constraint, but it is very difficult to obtain arbitrary sparsity levels outside a small interval.
>
> [Continues in Part 2]

---

> ### Author Response · Authors · 2018-11-19
> **Answer to reviewer (Part 2)**
>
> About the dataset sizes:
> ---------------------------------
> When comparing graph quality on real data, we usually face the challenge of not having in hand an actual ground truth graph. To have concrete quantitative comparisons, we chose the MNIST experiment as we could use classification error of label propagation as a proxy for graph quality. Note that for 60,000 nodes, saving a full adjacency matrix would need 28.8 GB of storage (for 64-bit floats, square form of W), therefore all non scalable algorithms (including Dong etal, Kalofolias, and Daitch) would run into problems.
>
> To add comparisons for larger data, as you suggested, we learned a large scale graph where we know the underlying node structure. We used 1000 signals from data from [Perraudin etal. 2018]. It consists of simulated cosmological mass maps, i.e. the amount of mass in the universe observed from earth in every direction. Hence  this data resides on the sphere, which can be considered as its underlying manifold. For this experiment, we learned a graph for a subpart of the sphere, i.e between 262,000 nodes (512x512 grid). For this new graph, we first see how well the theta bounds approximate the final graph sparsity we obtain. We then compute the first few eigenvectors of the Laplacian and see that the two first non trivial eigenvectors give an almost square 2-D grid embedding (like Laplacian eigenmaps), even if there was no information of x,y,z coordinates in our original data. We will add plots in the appendix (due to space constraints) showing the 2-D embedding we obtain. We note that the visual quality of the obtained embedding is significantly worse using A-NN and slightly worse using l2.
>
> In terms of time complexity, as we mentioned in Section 5.5, our algorithm can learn in reasonable time a graph of 1 million nodes on a desktop running Matlab. We extended slightly this experiment by adding a scalability figure in the appendix. While this is a proof of scalability, for really large data one will have to consider implementation details like memory management, and using a faster programming language (like C++).
>
>
> Locality Sensitive Hashing:
> ------------------------------------
>
> We are happy to cite LSH among other possibilities for finding A-NN. While a different A-NN technique might provide a better initial support and hence overall results, we believe that providing that A-NN is above some quality threshold, the final quality of the results will not be affected so much. Indeed the task of the optimization problem is to select which edges should be kept from the initial support (that is much larger than the final graph). So the optimization should compensate for at least some of the A-NN errors.
> Take for example Figure 4. To obtain the L2 and Log graphs of degree k=10, we started from an A-NN graph of degree k=30 (yellow line, up right) and set to zero many erroneous edges by learning.
>
> While studying the effect of different A-NNs to the final quality of the graph is interesting, this contribution focuses on scaling the problem of graph learning. Doing a fair and complete comparison with more A-NN models would be a long publication itself, and could make our submission lose its focus.
>
>
> Tightness of theta intervals with outliers in data:
> ---------------------------------------------------------------
> The theta intervals of equation (17) are indeed robust to outliers. We run experiments adding outliers (10% images contaminated with large amounts of Gaussian noise) in the MNIST dataset, and plotted the theoretical versus obtained sparsity of graphs versus the choice of parameter theta. The result is almost identical with the one of Figure 2b. The explanation is that the theta intervals of eq. (17) are really dependent on the smallest distances in $Z$, rather than the largest ones: $Z^\hat$ is sorted, and B as well. Adding outliers induces additional nodes distances larger than usually, that never make it in the equation of 17, except for the few outlier nodes.
>
> To complete the experiment, we also tried adding 10% duplicate images instead of outliers. In that case, we have 10% pairs of images that have essentially zero distances with each other. We thought this could affect the intervals much more, but we were wrong: the theta intervals are again very robust to this duplicates in the data: while the values of theta proposed are an order of magnitude larger in the case of duplicates, these values obtain graphs with degree within distance 1 from the desired one, as is the case for the outliers data, and the original MNIST data. We have added this new experiment in the appendix, but could also try to fit it in the main paper material.

---

### Author Response · Authors · 2018-11-19
**The reference of the spherical data we mentioned**

This is the reference to the paper [Perraudin etal 2018] where the spherical data were used for deep learning:

Nathanaël Perraudin, Michaël Defferrard, Tomasz Kacprzak, Raphael Sgier
DeepSphere: Efficient spherical Convolutional Neural Network with HEALPix sampling for cosmological applications

Link: https://arxiv.org/abs/1810.12186

---

### Author Response · Authors · 2018-11-27
**Summary of changes in new version**

Dear reviewers,

We would like to sincerely thank you for your constructive comments. After implementing them, we believe that the current version of the paper is not only stronger, but also clearer.

Our main focus was to strengthen the experimental section. The new version better illustrates  the advantages of large scale graph learning.

- For better clarity, we have re-organized the experiment by type : a) approximation quality of our scaled model compared to the original non-scaled version (section 5.1), b) quality of our automatic parameter selection (section 5.2), c) benefit from learning versus A-NN for large scale applications, i.e. assessment of the quality of the model (section 5.3, 5.4, 5.5), and d) scalability of the model (section 5.6)
- We added a comparison with Daitch “soft” and “hard” models both in terms of quality and scalability. To do so, we first implemented their original algorithms, but also proposed new, scalable versions based on our Section 3.
- We added a new experiment on larger graphs (262,144 nodes), where we focus both on time but also on graphs accuracy. We demonstrate how large-scale graph learning can recover a manifold, much better than usual A-NN graphs.
- We showed the accuracy of the theta parameter estimation for the new, larger dataset and also its robustness against outliers or duplicates (the latter part is in the appendix due to space limitations).

Eventually, we fixed other minor issues, including the addition of the missing reference and the clarification of some specific points.

We thank you again for your time and suggestions.

---

### Meta-Review · Area_Chair1 · 2018-12-16
**Probable accept based on majority vote.**

**Confidence:** 4
**Recommendation:** Accept (Poster)

**Metareview:**

The paper is proposed as probable accept based on current ratings with a majority accept (7,7,5).